# Bilateral Foot Orthoses Elicit Changes in Gait Kinematics of Adolescents with Down Syndrome with Flatfoot

**DOI:** 10.3390/ijerph17144994

**Published:** 2020-07-11

**Authors:** Daniele Galafate, Sanaz Pournajaf, Claudia Condoluci, Michela Goffredo, Gabriella Di Girolamo, Carlotta Maria Manzia, Leonardo Pellicciari, Marco Franceschini, Manuela Galli

**Affiliations:** 1IRCCS San Raffaele Pisana, 00163 Rome, Italy; daniele.galafate@sanraffaele.it (D.G.); claudia.condoluci@sanraffele.it (C.C.); michela.goffredo@sanraffaele.it (M.G.); gabriella.digirolamo@sanraffaele.it (G.D.G.); carlotta.manzia@sanraffaele.it (C.M.M.); leonardo.pellicciari@sanraffaele.it (L.P.); marco.franceschini@sanraffaele.it (M.F.); 2Department of Human Sciences and Promotion of the Quality of Life, San Raffaele University, 00166 Rome, Italy; 3Department of Electronic, Information and Bioengineering, Politecnico di Milano, 20133 Milan, Italy; manuela.galli@polimi.it

**Keywords:** down syndrome, flat foot, foot progression, GPS, GVS, gait analysis

## Abstract

*Background:* Subjects with Down Syndrome (DS) are characterized by specific physiological alterations, including musculoskeletal abnormalities. Flat Foot (FF), caused by hypotonia and ligament laxity, represents one of the most common disabling disorders in this population. Conservative treatments promote the use of orthopaedic insoles and plantar supports. The aim of this study was to evaluate the impact of Foot Orthoses (FOs) on the gait pattern of subjects with DS, assessing the biomechanical effects associated with their use. *Methods:* Twenty-nine subjects were screened under two conditions—walking barefoot (WB); with shoes and insoles (WSI), during three trials for each. Assessments were performed through the 3D gait analysis, using an optoelectronic system, force platforms, and video recording. Specifically, synthetic indices of gait kinematics, i.e., gait profile score (GPS) and gait variable score (GVS) were calculated and compared with Wilcoxon signed-rank test, to evaluate between-conditions. *Results:* Significant variations were found in GVS foot progression index, representative of foot rotation during walking, in adolescents only. *Conclusions:* Bilateral FOs has a positive immediate impact on gait quality in adolescents with DS, as confirmed by quantitative analysis. FOs prescription is an evidence-based early approach to slow down biomechanical abnormalities and prevent relative symptoms.

## 1. Introduction

In Down Syndrome (DS), alterations involving bones, muscles and joints, lead to movement and coordination impairments and determine an altered gait pattern [1]. Among clinical impairments observed in people with DS, Flat Foot (FF) represents one of the major orthopaedic deformities. FF is a structural and functional misalignment of the foot, with a prevalence of about 60% in the DS population. It was identified as one of the reasons for disability in these subjects [2]. This postural deformity could be caused by low muscle tone (hypotonia), low ligament tension (ligamentous laxity) and joint hyper flexibility [3].

Walking impairments and typical gait patterns are widely assessed in DS with and without FF, as compared to healthy people, through gait analysis. Nowadays, many innovative systems are used in the clinical routine to quantify the functional performance of subjects, from children to elderly people. In this context, gait analysis is widely applied, from targeted clinical decision making to quantitative monitoring of post-treatment changes [4]. For this reason, the literature is concerned about innovative protocols and processing techniques to synthetically describe gait data and thus improve the interpretations [5,6]. A study by Galli et al. [7] analysed gait biomechanics of 98 children with DS, compared to 30 healthy ones. Specifically, subjects with DS walked with a greater hip flexion during the whole gait cycle, a higher knee flexion in the stance phase, a limitation of the knee range of motion, and a smaller plantarflexion of the ankle at initial contact. The gait pattern of subjects with DS was also compared with their healthy peers in terms of the gait variable scores (GVSs) and gait profile score (GPS), showing significant differences in all lower limb joints, at all planes of movement [8].

Literature on foot and ankle biomechanics in subjects with DS and FF has evidenced a relationship between the degree of FF deformity and several distal kinematic and kinetic inefficiencies. The most relevant ones concern reduced ankle joint, plantarflexion moment, and peak of power in push-off, with a global propulsive deficit in walking, and a modified pattern of centre of pressure [9,10]. A study quantitatively evaluated the relationship between FF and walking changes in children with DS [11]. The results showed that subjects with FF expressed a worse functional gait pattern than their peers without FF, in terms of ankle kinetics. This feature was also highlighted by another study [12], which assessed the gait pattern of subjects with DS and their FF condition. In particular, the authors investigated the correlation between the entity of the plantar arch (which determines the grade of FF) and the external rotation of the foot during gait. They revealed how FF can cause abnormal external rotation over the walking stages. Different motor strategies were observed in subjects with DS, compared to healthy ones. Specifically, a reduced range of motion and stiffness in the proximal joints were reported as possible compensatory mechanisms to muscle weakness [7,13].

Proportional to the grade of FF, standard conservative treatments involve the use of foot orthoses (FOs) to support, align and correct the foot, and the joints of the lower limbs. The recommendation to use FOs is often linked to their feasibility (i.e., easy costume and their non-invasiveness), especially considering its common use among young subjects. The literature on the effects of FOs is controversial. It is frequently based on podoscopic examinations, during the upright orthostatic position [14,15].

Several studies investigated changes in gait biomechanics, due to FOs in healthy subjects [15,16,17]. A recent study was conducted to assess the effects of special footbed on children suffering from FF. In particular, this investigation determined the short-term benefits of personalized arch support [16]. Additionally, Kulcu et al. evaluated immediate changes in gait with the use of bilateral silicone insoles, hypothesizing that silicone insoles would have improved joint kinematics and kinetics [17].

In this context, to the best of our knowledge, a limited number of studies have involved subjects with DS [18,19]. A pilot study by Selby-Silverstein et al. [18] compared the gait of children with DS wearing sneakers, with and without FOs, to the gait of healthy children. They found an immediate effect of FOs on decreasing heel eversion in quiet standing, in the group of children with DS. While during gait of the same group, FOs caused a more internally rotated transverse plane foot angle, decreased trial-to-trial variability of foot function parameters and walking speed, and increased trial-to-trial variability of ankle moment.

The use of FOs in children with DS is escalating. However, their efficacy in terms of changes in gait biomechanics still requires more detailed investigations. This paper aimed to address the lack of studies on this topic, in order to translate the quantitative analysis in evidence-based clinical and rehabilitation practice [20,21].

Therefore, the objective of this study was to test the immediate effects of using the FOs on the gait pattern of subjects with DS, suffering the FF condition. It was hypothesised that the FOs would shift the lower limb joints’ kinematics toward a more physiological model.

## 2. Materials and Methods 

### 2.1. Participants

An observational retrospective cross-sectional study was conducted on a database of subjects with DS at our Institute for Scientific Research and Health Care (Rome, Italy), between 2012 and 2017.

Inclusion criteria for the patient selection were—presence of trisomy 21; normal vision and hearing; presence of bilateral FF of II or III grade; presence of FOs for FF (custom made of the same orthosis and prosthesis centre); ability to walk unassisted for at least 10 metres; and absence of orthopaedic or cardiovascular comorbidity affecting gait. Exclusion criteria were—presence of other severe medical conditions; inability to understand or execute the task; and inability to provide informed consent.

### 2.2. Ethical Aspects

Since March 2012, the Italian Data Protection Authority (Garante per la protezione dei dati personali) declared that the IRCCSs (Istituto di Ricovero e Cura a Carattere Scientifico — Institute for Scientific Research and Health Care) can perform retrospective studies, without the approval of the local Ethical Committee [22], and only a formal communication is needed. Such communication was registered by the local Ethical Committee (date: 20/12/2019; code number RP 19/35), which waived the need for participants’ consent.

However, all subjects or subjects’ parents (in case of under 18 y.o.), as required by the institutional policy routines, should be informed about the acquisition protocol and should sign a consent to the use of anonymous data, before performing the gait analysis. Subjects whose consent was missing for any reason were excluded.

Each record in the database was identified through a unique alphanumeric code, in order to preserve the patient’s privacy. The study complied with the Declaration of Helsinki.

### 2.3. Procedures

The following demographic data were extracted from the electronic medical records—age, gender, weight, height, body mass index, and intelligence quotient.

All subjects conducted 3D gait analysis under two different test conditions—walking barefoot (WB); and walking with shoes and FOs insoles (WSI). The custom-made FOs were supplied by the same orthoses and prosthesis centre for each patient. The provided FOs was made of thermo-formable polymeric synthetic material, equipped with the plantar arch support and an enveloping rear foot structure, in order to maximize the heel stability. The longitudinal height of the insoles’ midfoot arch was set on the basis of foot flatness grade.

The gait analysis was conducted with the following equipment—a stereophotogrammetric system (SMART-DX, BTS Bioengineering, Milan, Italy) composed of 8 infrared cameras and reflective markers for the kinematics assessment; 4 force platforms (Kistler, CH) for measuring the kinetics (i.e. ground reaction forces); and 2 RGB video cameras (BTS Bioengineering, Milan, Italy) for video recording. All systems acquired the data synchronously, with the following sampling frequencies—SMART-DX: 100 Hz; Kistler: 2000 Hz. The accuracy of the stereophotogrammetric system was <0.1 mm for a calibration volume of 1.60 × 1.80 × 2.0 [m] (width × height × depth, respectively).

The reflective markers were placed with adhesive tape on the skin of the subjects, in correspondence of the anatomical points defined by the Davis Heel protocol for gait analysis [23]. The anthropometric data were measured, as suggested in the literature [23]. The motion analysis consisted of two phases for each condition: (i) a standing phase; (ii) a walking phase. During the standing phase, the subject conducted a 5-second-long postural task standing upright on the force platforms with arms along the body and opened eyes. During the walking phase, subjects walked at a self-paced comfortable speed, over a 10-meters path, crossing through the force platforms. Four walking trials were performed under every condition, in order to guarantee data consistency. A 10-minutes rest was allowed between conditions to minimize fatigue [24] and a trial was considered as acceptable when the subjects walked with no visible abnormalities in the process.

The acquired data were analysed, and the spatiotemporal parameters and lower limb joints’ kinematics were calculated. The ground reaction forces were gathered but these were not presented or discussed in this paper. Data analysis was conducted with the Smart Analyser software (BTS Bioengineering, Milan, Italy), which allowed to segment each gait cycle and to calculate the following spatio-temporal parameters:gait cycle (s)—mean temporal duration of the gait cycle that begins with initial heel contact and ends with the subsequent heel contact of the same limb;% stance (as a % of the gait cycle)—% of the gait cycle that begins with initial contact and ends at toe-off of the same limb;% double support (as a % of the gait cycle)—% of the gait cycle during which the feet are placed on the ground;mean velocity (m/s)—the mean velocity of progression for each limb;stride length (m)—distance between successive ground contacts of the same foot;step length (m)—longitudinal distance from one-foot strike to the next one;step width (m)—mediolateral distance between the two feet during double support.

The joint kinematics were normalized as a percentage of the gait cycle, producing sagittal kinematic plots of the pelvis, hip, knee, and ankle, for each cycle.

The Gait Variable Scores (GVSs) were calculated, as proposed by Baker et al. [25]. Specifically, the GVSs of the following joints were computed for both sides under the two walking test conditions, in order to assess the joint kinematic effects, due to the use of FOs during gait—pelvic obliquity; pelvic tilt; pelvic rotation; hip adduction/abduction; hip flexion/extension; hip rotation; knee flexion/extension; ankle dorsi/plantarflexion; and foot progression.

Moreover, from the GVSs data, the Gait Profile Score (GPS) was computed, which summarized the overall deviation of the joint kinematics from normative data [25]—the smaller the GPS values, the more physiological the gait pattern.

### 2.4. Statistical Analysis

All previously defined parameters were calculated for each participant in each walking condition. The mean values over the four walking trials were calculated.

The patients were divided with a criterion based on age. Specifically, patients were divided into two groups—adolescents (age < 18) and adult (age ≥ 18). Descriptive statistics were used to describe the demographic and clinical characteristics of the groups; mean ± standard deviation (SD), and frequency with relative percentage were computed for the ordinal and categorical variables, respectively. The Kolmogorov–Smirnov test were run preliminary to assess the normality of the data; an acceptable range indicating the data normality was considered for significance values >0.05 and for a skewness lying between –1 and +1 [26]. As the data were presented with a non-normal distribution, the Wilcoxon signed-rank test was performed to compare the spatial and temporal parameters between the two different walking conditions. The statistical analysis was conducted though Matlab^®^ (MATLAB and Statistics Toolbox Release 2012b, The MathWorks, Inc., Natick, Massachusetts, United States) and the statistical significance was set for *p* ≤0.05.

## 3. Results

Twelve adolescents (mean ± SD age: 13.8 ± 2.6 years [range 9–17 years]; 41.7% male) and seventeen adults (mean ± SD age: 26.9 ± 8.3 years [range 18–48 years]; 64.7% male) with DS were enrolled in this study. Figure 1 reports the study procedure flow. The demographics and clinical characteristics of the sample are presented in Table 1. All subjects tolerated the FOs well; no adverse events were extracted from the clinical and electronic records.

Table 2 depicts the spatio-temporal parameters obtained from the kinematic gait analysis, in the two different walking test conditions. The results of the Wilcoxon signed-rank test showed no significant differences in all spatio-temporal gait parameter measurements, obtained under the two different walking conditions for the adolescents and adults.

Table 3 depicts the GVSs for each joint angle, and the GPS under the two walking conditions. The analysis of GVSs showed a significant change of foot progression when adolescent walked with shoes and FOs (WSI) in both the right (Z = 2.670, *p*-value = 0.008) and left (Z = 2.209, *p*-value = 0.027) side; this difference was not found in adults (*p* > 0.05). No significant differences were obtained in other GVSs parameters and in GPS for adolescent and adult samples (*p*-value > 0.05).

## 4. Discussion

In subjects with DS, the FOs are the most common solution for FF. However, limited scientific literature has quantitatively analysed the effects of FOs in this population [18,19]. The present paper aimed to compensate the lack of scientific literature in this field, presenting the results of a retrospective study on the immediate effects of bilateral FOs on gait kinematics, in subjects with DS and FF.

The gait kinematics of twenty-nine subjects with DS and FF was collected. Data of adolescents (N = 12) and adults (N = 17), who walked barefoot (WB) and with FOs (WSI), was analysed separately. The FOs were custom-made and tailored to each subject’s flatfoot grade.

The obtained spatiotemporal gait parameters represent the typical ambulation of the DS population described in the literature [7,8]. The comparison between the two walking conditions did not reveal any significant differences in the adolescents or in the adults, in terms of gait cycle time, stance and double support phases, mean velocity, stride and step lengths, and step width. Our findings were in accordance with Kulcu et al. [17], who did not find any immediate effects of bilateral silicone insoles in spatiotemporal gait parameters in healthy adults with FF. Increasing spatiotemporal parameters such as gait speed, was not the main aim of the FOs prescription for subjects with FF. They were, in fact, addressed to support the foot and ankle stability, thus promoting the walking quality by restoring all lower limbs’ joint misalignments and correcting the motor pattern alterations.

The use of GPS and GVS was proposed to describe the gait pattern in many pathologies [27,28], but they were not sufficiently applied for the evaluation of subjects with DS with FF. In the present study, the obtained joint angles represented by the GVS and GPS, in comparison to the typical values of healthy people, showed interesting variations elicited by FOs. Specifically, the use of FOs significantly decreased the right and left GVS foot progression score in the adolescent (*p* < 0.05). This outcome indicated that the subjects in this condition walked with more aligned physiological feet. On the other hand, the adult subjects did not register any significant variations. The other GVSs and the summary measure GPS did not change when the subjects used FOs.

The foot progression angle was defined as the angle between the line from the calcaneus to the second metatarsal, and the line of progression averaged from the heel strike to toe off, during the stance phase of walking for each step. This index represented the rotation of the foot in horizontal plane (Figure 2).

Therefore, a small GVS foot progression meant a smaller external rotation during the stance phase, hence a greater foot stability [29]. In other words, the plantar external rotation during the progression of the foot, which is recognized as a common characteristic among subjects with DS [8,12], decreased due to the use of shoes and FOs. These results were in line with Selby-Silverstein et al. who found an immediate effect of the FOs in children with DS, in terms of decreasing heel eversion during gait, caused by a more internal rotation of the transverse plane foot angle [18]. This finding was more specifically explained by the biomechanics of ankle joints and the pressure distribution during the stance phase [29], and by the anatomy of FF [30]. In fact, a decreased heel eversion (near neutral version) during the plantarflexion and ankle dorsiflexion reduced the high medial pressure load, thus resulting in a better planar pressure distribution. Therefore, a more powered push-off, even slightly, would be achieved at the pre-swing phase, which was shown by Galli et al. [7] to be decreased in DS subjects, compared to the healthy ones. This, in turn, might result in a well-adapted walking with a greater stability [31] in terms of smaller bilateral GVS of foot progression. Evidently, sophisticated pressure map systems (i.e., sensorised insoles) available for podiatric diagnosis, could confirm the change of plantar pressure load in detail, following the use of FOs and during the ecological walking.

Of course, the retrospective design of this study did not allow to collect sufficient specific clinical data. These detailed characteristics would help to better understand who might benefit more from the use of FOs. Another limitation, which deserve further investigations, was the lack of a walking test condition with shoes and without FOs. Additional research is needed on analysis of data extracted from platforms for the study of joint dynamics. Therefore, future studies should detect potential relation between biomechanical and clinical data, taking into account the effect of this non-invasive approach on activity, participation, and quality of life.

From a clinical perspective, our results suggest that the use of FOs might be more beneficial in adolescents than in adults. However, our data cannot support the effect of a prolonged application of FOs. Perhaps the complex motor skills might take longer to be improved, and further changes could be found if data was collected during follow-up. The research agenda should concern the effect of long-term use of FOs in subjects with DS.

## 5. Conclusions

The prescription of custom-made FOs is a conservative early approach to manage FF in subjects with DS. It is safe and low-cost and elicits an immediate positive effect on foot progression angle, thus, improving gait quality in adolescents with DS. However, FOs might be complementary tools to slow down the biomechanical abnormalities and the relative symptoms. All DS postural disorders and motor impairments call for a tailored rehabilitation program among conservative approaches.

## Figures and Tables

**Figure 1 ijerph-17-04994-f001:**
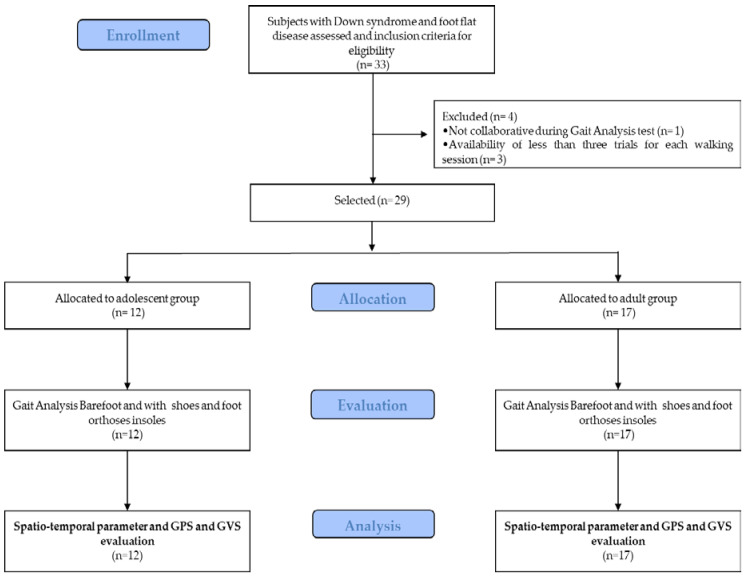
Flowchart of the experimental procedures.

**Figure 2 ijerph-17-04994-f002:**
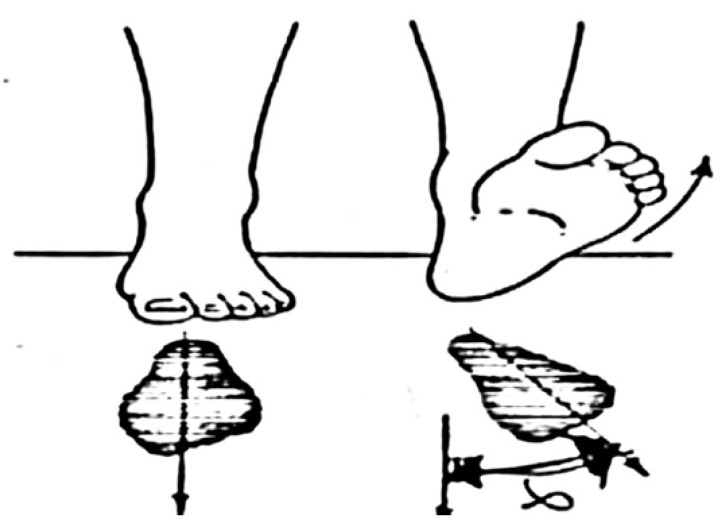
Foot progression representing the rotation of the foot in horizontal plane.

**Table 1 ijerph-17-04994-t001:** Demographic and clinical characteristics of the adolescents and adults.

Variable	Adolescents (N = 12)	Adults (N = 17)
Age (years)	13.8 ± 2.6	26.9 ± 8.3
Gender (Male/Female)	5 (41.7%)/7 (58.3%)	11 (64.7%)/6 (353%)
Weight (kg)	50.0 ± 13.7	59.3 ± 14.2
Height (cm)	142.5 ± 10.9	148.3 ± 9.0
Body mass index (kg/m^2^)	24.5 ± 5.1	26.7 ± 4.4
Intelligence Quotient	67.0 ± 11.4	69.1 ± 9.3

Notes: Data are reported as mean ± SD or frequency with relative percentage.

**Table 2 ijerph-17-04994-t002:** Spatio-temporal gait parameters for adolescents and adults under the two walking conditions—walking barefoot (WB) and walking with shoes and FOs insoles (WSI).

Parameters	Adolescents (N = 12)			Adults (N = 17)		
	WB	WSI	*p*-Value	WB	WSI	*p*-Value
Right gait cycle (s)	1.14 ± 0.16	1.17 ± 0.11	0.295	1.26 ± 0.16	1.27 ± 0.14	0.743
Left gait cycle (s)	1.14 ± 0.15	1.16 ± 0.1	0.442	1.26 ± 0.15	1.28 ± 0.15	0.783
Right % stance (as a % of the gait cycle)	62.95 ± 4.12	64.41 ± 3.45	0.372	62.27 ± 3.62	63.54 ± 2.83	0.148
Left % stance (as a % of the gait cycle)	63.20 ± 4.69	63.34 ± 2.97	0.735	62.44 ± 1.79	64.02 ± 2.91	0.068
Right % double support (as a % of the gait cycle)	12.83 ± 3.59	13.74 ± 3.39	0.644	12.50 ± 2.54	14.07 ± 2.50	0.085
Left % double support (as a % of the gait cycle)	13.00 ± 4.61	13.87 ± 2.95	0.518	12.56 ± 2.38	13.9 ± 2.60	0.179
Mean velocity (m/s)	0.75 ± 0.22	0.83 ± 0.15	0.265	0.73 ± 0.20	0.79 ± 0.22	0.436
Right stride length (m)	0.85 ± 0.18	0.97 ± 0.14	0.060	0.89 ± 0.18	0.99 ± 0.22	0.202
Left stride length (m)	0.84 ± 0.18	0.95 ± 0.16	0.116	0.88 ± 0.18	0.98 ± 0.21	0.129
Right step length (m)	0.42 ± 0.09	0.48 ± 0.08	0.123	0.44 ± 0.10	0.49 ± 0.10	0.185
Left step length (m)	0.42 ± 0.10	0.49 ± 0.08	0.069	0.44 ± 0.08	0.49 ± 0.11	0.196
Step width (m)	0.15 ± 0.06	0.14 ± 0.06	0.711	0.15 ± 0.04	0.15 ± 0.06	0.850

Abbreviations: WB, Walking Barefoot; WSI, Walking with Shoes and FOs Insoles. Note: In bold are the significant *p*-values. Data are reported as mean ± SD.

**Table 3 ijerph-17-04994-t003:** Gait variable score (GVS) and gait profile score (GPS) for adolescents and adults under the two walking conditions—walking barefoot (WB); and walking with shoes and FOs insoles (WSI).

Parameters	Adolescents (N = 12)			Adults (N = 17)		
	WB	WSI	*p*-Value	WB	WSI	*p*-Value
Right GVS pelvic obliquity (°)	3.4 ± 1.15	3.18 ± 1.28	0.552	2.59 ± 1.33	2.56 ± 1.03	0.945
Left GVS pelvic obliquity (°)	3.38 ± 1.30	3.47 ± 1.47	0.767	2.84 ± 1.09	2.6 ± 0.75	0.918
Right GVS pelvic tilt (°)	4.03 ± 3.01	5.46 ± 2.88	0.138	4.89 ± 3.57	5.32 ± 3.54	0.654
Left GVS pelvic tilt (°)	4.10 ± 3.06	5.60 ± 2.91	0.187	5.1 ± 3.41	5.37 ± 3.41	0.809
Right GVS pelvic rotation (°)	5.04 ± 1.38	4.88 ± 1.18	0.869	3.89 ± 1.24	4.83 ± 1.31	0.060
Left GVS pelvic rotation (°)	5.16 ± 2.04	5.74 ± 1.98	0.598	3.84 ± 1.14	4.63 ± 1.15	0.058
Right GVS hip adduction/abduction (°)	5.62 ± 2.70	5.72 ± 2.96	0.947	4.57 ± 3.00	5.00 ± 3.44	0.480
Left GVS hip adduction/abduction (°)	6.84 ± 3.23	7.27 ± 3.56	0.921	4.11 ± 2.28	5.16 ± 3.55	0.293
Right GVS hip flexion/extension (°)	9.25 ± 4.62	9.67 ± 5.44	1.000	9.31 ± 4.42	9.60 ± 4.28	0.904
Left GVS hip flexion/extension (°)	9.92 ± 4.26	10.06 ± 5.12	0.921	9.06 ± 3.44	9.08 ± 4.50	0.718
Right GVS hip rotation (°)	11.27 ± 6.5	10.64 ± 4.61	1.000	14.53 ± 11.43	14.31 ± 10.98	0.836
Left GVS hip rotation (°)	15.84 ± 6.97	17.5 ± 10.38	0.817	12.29 ± 7.06	11.91 ± 7.76	0.986
Right GVS knee flexion/extension (°)	10.81 ± 5.6	10.65 ± 3.87	0.817	12.93 ± 5.42	12.25 ± 5.48	0.642
Left GVS knee flexion/extension (°)	12.55 ± 5.49	10.74 ± 3.37	0.510	12.55 ± 5.86	11.35 ± 6.34	0.502
Right GVS ankle dorsi/plantarflexion (°)	8.50 ± 3.43	9.15 ± 2.86	0.644	6.74 ± 1.71	7.37 ± 2.34	0.642
Left GVS ankle dorsi/plantarflexion (°)	8.08 ± 3.08	8.82 ± 3.56	0.531	6.74 ± 2.04	7.87 ± 2.85	0.191
Right GVS foot progression (°)	17.75 ± 6.19	9.98 ± 3.69	0.008	14.3 ± 7.13	10.99 ± 6.19	0.191
Left GVS foot progression (°)	19.20 ± 8.06	11.71 ± 4.86	0.027	15.34 ± 7.84	10.81 ± 6.06	0.063
Right GPS (°)	10.17 ± 1.86	9.03 ± 1.63	0.124	10.21 ± 3.35	9.64 ± 3.43	0.705
Left GPS (°)	11.49 ± 2.26	10.65 ± 2.17	0.207	9.63 ± 3.02	9.07 ± 2.71	0.558

Abbreviations: GPS, Gait Profile Score; GVS, Gait Variable Score; WB, Walking Barefoot; WSI, Walking with Shoes and FOs Insoles. Note: In bold are the significant *p*-values. Data are reported as mean ± SD.

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
