# Peer review of "Bilateral Foot Orthoses Elicit Changes in Gait Kinematics of Adolescents with Down Syndrome with Flatfoot"

_ijerph, 2020, doi:10.3390/ijerph17144994_

Round 1

Reviewer 1 Report

This study investigates the immediate effect of foot orthosis (FO) on people with Dawn syndrome (DS) having flat foot (FF). They compared spatio-temporal parameters and kinematics using GPS and GVS in one gait cycle. In this context, this work is interesting and could tell something new to the readers. The work flow is acceptable and the results are well presented.

However, there are some concerns that should be discussed.

  1. Although the authors mentioned the FF as a common disability in people with DS but still it is not clear why they want to study the FF particularly in the DS and what are they motivations. What does make The FF different in the DS comparing to the other patients? Do people with DS using different compensatory strategy? What is missing here is the comparison between the FF in the DS and the other group with FF and elaboration of gait strategy in the DS with the FF and how it differs with the other patients. To be honest, I don’t see any strong reasons results or conclusion here in the text to support studying the FF in the DS specifically.
  2. The methodology of classifications is not clear. First, the age range includes adolescence and adult subjects. there is no distinguish between them and not even mention about it in discussion. It would be interesting to see if there are differences between these two subgroups. Second is the flat foot grade. If the FOs are costumed according to the flat foot grade (what is flat foot grade?) then why the results are separated in the right and left sides? What are the author’s reasons to separate left and right sides? What is the relationship between the flat foot grade and the significant parameters?

Minor issues

  1. I strongly recommend to review the whole text grammar wise. In order to understand the sentences better, please break down the long ones into the shorter ones.
  2. The number of citations is too high for the work in this level. Even the future work and conclusion are cited. It brings this issue to the mind that authors don’t have anything say by themselves. Higher number of citations doesn’t guarantee more qualified research.

Abstract

L.22: “(3) Results: Significant variations were found in GVS values, especially the foot progression index, representative of foot rotation during walking.”

This is not true. You find GVS significant differences ONLY in foot progression angle (FPA).

Introduction

Lines 30-81 of the introduction require a major review in grammar and shortening the sentences.

L.30:” In Down Syndrome (DS), alterations involving bones, muscles and joints lead to movement and

coordination complications and determine an altered motor pattern during gait

Please rephrase this sentence. by coordination you mean coordinates? Motor pattern? (gait pattern and motor control?). The whole sentence, as if I understood correctly, suggests that the change in musculoskeletal structure of patients with DS alters the gait patterns and consequently motor control is it true or it is more mutual relationship?

L.35: a sentence better not to be finished with a verb.

L.36 Large study? please find a better adjective.

L.43: greater variability in (more changes)

L.51: extra dot at the end.

L.54: referring as

L.57 “the standard conservative treatments are based”.

No need to use ARE here.

L.57 “the clinicians proposed different treatments: the standard conservative treatments are based on the use of Foot Orthoses (FOs), which are used for the support, alignment and correction of the foot and joints of the lower limbs. “

It is supposed after using “:” you introduce or name several (two or three) methods but “the standard … “is an independent sentence from the one before “:”, please rephrase the whole sentence.

L61-62: Are you talking about feasibility?

L67: To the best of our knowledge

L69-71: “… children with DS showing that subjects with FF expressed a worst functional gait pattern in terms of ...”

Please rephrase the lines and use worse instead of worst.

Materials and methods

  1. 100: Please mention the participant numbers. Is there more information available about the age interval? Did authors consider the puberty growth into account? How many of the patients are under 18?

L.120: Please somewhere here refer to table 1.

L.127: In this text, the term “flat foot grade” or “foot flatness grade” has been repeated several times. It would be interest of the readers if authors provide some information about it.

L.130: (width, height and depth respectively)  

L.150-155: I think these lines (gait cycle, stance and double support phases) may not be necessary.

Results

Why the results are presented in left and right sides?

L.205: “The analysis of GVSs showed a significant change of foot kinematics when subjects walked with shoes and FOs (WSI) in both right (Z=3.110, p-value=0.002) and left (Z=3.035, p-value=0.002) side”.

This sentence implies that all the foot kinematics have significance differences.

Discussion

The discussion is very long and messy. The number of references in the discussion are almost near to the ones in the introduction. Also, some parts of it could or should be used directly in the introduction. Some parts of discussion do not seem to be relevant to this study.

L.220: As I mentioned before the authors should distinguish between adolescents and adults or at least discuss about it. As there is not proper information, regarding the subject’s age, available by authors I think this is one of the rather weak point of this study.

L.223: “…alignment and correction of the foot while serving a more physiological load on the heel [13].”

This sentence is repeated in introduction.

L.225-229: “…In this context, walking velocity, not only does not represent an issue to cover but also it has been shown that an increased walking speed indicates the lack of dynamic stability

What does “…not represent an issue to cover” means?

Also, it seems this is in contradiction with the Line 51 of the introduction where the gait velocity is identified as a prevalent impairment.

L.232-236: “Also, Kulcu et al. evaluated immediate changes in gait with the use of bilateral silicone insoles, hypothesizing that silicone insoles would have improved joint kinematics and kinetics [42]. Although, their study was conducted in adults with flexible FF, they did not find any beneficial immediate effects of temporal parameters (walking velocity, step length, step time), accordingly with our results.”

  1. some part of this could be written in introduction and referred here.
  2. as far as table 1 presents, there are significant differences for your step length results, despite of the Kulcu study, so I see contradiction in your claim and Kulcu’s results.

L241-244: “Double support time and step width have been shown to be extremely correlated with balance:

Do you have any results from your data to support this claim?

their increase in young people with DS is due to adequate adaptations to provide the greater than normal stability needed to control comfortable walking, thus to adjust their body centre of pressure, similar to their peers with Typical Development (TD) chronologically older.”

The part is written in a confusing way. Please write in shorter sentences.

  1. 247-249: These sentences are unclear.

L.260: probably wrong reference 35 instead of 34 (however the same repetition in material and method)

L.264-267: this definition of FPA better goes to the material and method.

L.270: what is Extra-rotation? You mean external rotation?

L.273-285: “Accordingly, Selby-Silverstein et al. has also found an immediate effect of the FOs in … of plantar pressure load following the use of FOs.”

As I understood the authors try to show smaller PFA leads to less external rotation and hence improvement the gait stability. The whole paragraph appears to be a discussion about Selby.Silverstein study and off from the current work. Do you have any results from your data to support this discussion part (better plantar pressure distribution?).

L.298: I don’t see why your future work also requiring a citation?

L.288: “A recent study was conducted to assess … of personalized arch support [48].

This I suggest to bring this sentence in introduction as a part of the motivation to conduct the current study not in the limitations.

Conclusion

This is not a proper conclusion. The conclusion supposed to be your final statement about the work not referencing the other works.

The results show only improvement in the FPA (GVS) and step and stride length so it would not fully support your hypothesis and authors may mention this.

Reviewer 2 Report

Well written manuscript which collects data from a small scale pilot to prove the utility of the FOs. The gathered data are very complete and may lead to very interesting conclusion. From a biomechanical  point of view, table 3 could be used to implement a scheletric model capable to investigate (and assess) the gait behaviour (for example extrrotation in Fig 2 could be better related to hip extra rotations and ankle joint variables). This reviewer Encourage both the publication of the paper and a more computational-predictive approach also based on antrophomorphic robotic model to increase the comprension of the phenomenon and consequently, assess the personalisation in the design of the orteses, as a future study.

Round 2

Reviewer 1 Report

Thanks to the authors and their efforts in order to reply properly to the comments. They almost relieved all my concerns and provided a better-quality version of the paper.

 However, some minor comments still are remained though I don’t need direct reply from the authors. It is just some suggestions and hope they could help to improve the paper.

  1. In the title you’ve added adolescence. Shouldn’t you add adolescence and adults?
  2. Having significant differences in the GVS progression angle for the adolescences comparing to the adults is an interesting finding and authors properly addressed the relation between FPA and foot external rotation. However, it would be great if the authors mention their ideas or hypothesis that what is the possible reasons for these differences in adolescences and how they may overcome it in adults.
  3. Thanks for clearing the presenting of the right and left sides data but the issue is that you don’t bring information in the paper unless you want to elaborate and discuss about it. the data for left and right sides intuitively bring some questions about why they are there and what would be authors’ explanations (i.e. is there any pathological or anatomical differences between left and right sides? or is there any issue about balance and symmetry to be mentioned?). So, I suggest either add few lines about right and left sides presenting purpose or present only the averaged left and right sides which help to read the results easier and understand them better.